

# A low dose of an organophosphate insecticide causes dysbiosis and sex-dependent responses in the intestinal microbiota of the Japanese quail (*Coturnix japonica*)

Eduardo Crisol-Martínez[1,6], Laura T. Moreno-Moyano[2], Ngare Wilkinson[1,3,4], Tanka Prasai[1,3], Philip H. Brown[1,3], Robert J. Moore[4,5] and Dragana Stanley[1,3,4]

[1] School of Medical and Applied Sciences, Central Queensland University, Rockhampton, Queensland, Australia
[2] School of Biosciences, University of Melbourne, Parkville, Victoria, Australia
[3] Institute for Future Farming Systems, Central Queensland University, Rockhampton, Queensland, Australia
[4] Poultry Cooperative Research Centre, University of New England, Armidale, New South Wales, Australia
[5] School of Science, RMIT University, Bundoora, Victoria, Australia
[6] Current affiliation: Central Queensland University, Melbourne, Victoria, Australia

Corresponding author
Eduardo Crisol-Martínez, eduardocrisol@gmail.com

## ABSTRACT

Organophosphate insecticides have been directly or indirectly implicated in avian populations declining worldwide. Birds in agricultural environments are commonly exposed to these insecticides, mainly through ingestion of invertebrates after insecticide application. Despite insecticide exposure in birds occurring mostly by ingestion, the impact of organophosphates on the avian digestive system has been poorly researched. In this work we used the Japanese quail (*Coturnix japonica*) as an avian model to study short-term microbial community responses to a single dose of trichlorfon at low concentration in three sample origins of the gastrointestinal tract (GIT): caecum, large intestine and faeces. Using next-generation sequencing of 16S rRNA gene amplicons as bacterial markers, the study showed that ingestion of insecticide caused significant changes in the GIT microbiome. Specifically, microbiota composition and diversity differed between treated and untreated quail. Insecticide-associated responses in the caecum showed differences between sexes which did not occur with the other sample types. In caecal microbiota, only treated females showed significant shifts in a number of genera within the Lachnospiraceae and the Enterobacteriaceae families. The major responses in the large intestine were a significant reduction in the genus *Lactobacillus* and increases in abundance of a number of Proteobacteria genera. All microbial shifts in faeces occurred in phylotypes that were represented at low relative abundances. In general, changes in microbiota possibly resulted from contrasting responses towards the insecticide, either positive (e.g., biodegrading bacteria) or negative (e.g., insecticide-susceptible bacteria). This study demonstrates the significant impact that organophosphate insecticides have on the avian gut microbiota; showing that a single small dose of trichlorfon caused dysbiosis in the GIT of the Japanese quail. Further research is necessary to understand the implications on birds' health, especially in females.

## INTRODUCTION

Organophosphates are the most widely applied insecticides (*Bondarenko et al., 2004*; *Zamy, Mazellier & Legube, 2004*), and they primarily act by inhibiting cholinesterase, a nervous system enzyme (*Marrs, 1993*). Avian exposure to these contaminants occurs through dermal contact, maternal transfer, inhalation and ingestion (*Smith et al., 2007*). Among all of these routes, ingestion is the predominant pathway of avian exposure, which can occur through the consumption of prey items such as insects that have been previously exposed and/or killed by insecticides (*Golden & Rattner, 2003*). Several studies have reported how organophosphate insecticides caused the mass death of birds (e.g., *Fleischli et al., 2004*; *Frank et al., 1991*), some of these through ingestion of contaminated prey (e.g., *Goldstein et al., 1999*; *Goldstein et al., 1996*). While these lethal responses to insecticide ingestion highlight toxicity to birds, few studies have examined effects of ingestion of sub-lethal organophosphate dosage on the gastrointestinal tract which may lead to chronic effects on bird health.

Trichlorfon is an organophosphate insecticide, which has been used since the 1950s in a wide number of systems, including tree orchards, vines, fruits, vegetables, field crops, pastures, forests, and turf (*De Oliveira, Moreira & Goes, 2002*). Trichlorfon is considered moderately toxic to birds (e.g., LC50 values in bobwhite quail (*Colinus virginianus*) = 720 ppm; in Japanese quail (*Coturnix japonica*) = 1,901 ppm) (*Hill et al., 1975*). Most of the toxicological research on trichlorfon in avian species has focused on its neurotoxicity; generally shown as severe decreases in anticholinesterase activity (e.g., *Hill, 1982*; *Iko, Archuleta & Knopf, 2003*). The majority of the toxicological assessments have used relatively high insecticide concentrations derived from general measures of toxicity such as the lethal dose or lethal concentration values of the species tested (e.g., *De Oliveira, Moreira & Goes, 2002*; *Slott & Ecobichon, 1984*). However, little work has assessed the toxicological impacts of insecticides using insecticide concentrations representative of likely exposure levels in the field; such as residues present at a given time after their application (e.g., *Joly et al., 2013*). Residue levels in pesticide-exposed items can be measured to calculate how much of the contaminant is ingested by birds to then understand the context of impact assessment (*Smith et al., 2007*).

Although the digestive system is the first contact point for contaminants, few studies have aimed to understand the toxicological impacts of pesticides on the gastrointestinal tract (GIT) (*Joly et al., 2013*). The microbiota present throughout the GIT is essential in development and maturation of the immune system (*Kelly & Conway, 2005*; *Umesaki et al., 1999*). The GIT microbiota contributes to digestive and fermentative processes, and prevents colonization by pathogens (*Kamada et al., 2013*), ultimately contributing to the energy and nutrient supply, immunity, and general well-being of the host (*Stanley, Hughes & Moore, 2014*). The importance that host-microbiome symbiotic relationships have in the success of animals is only recently being recognized by evolutionary biologists

and ecologists (*McFall-Ngai et al., 2013*). However, environmental stresses can shift these relationships, thus affecting host performance and/or health (*Myers, 2004*).

To date, most of the studies characterising the GIT microbiota of birds have been carried out in chickens (e.g., *Dumonceaux et al., 2006*; *Stanley et al., 2013*). Only a few other studies have characterised the microbiota of other bird species, including kakapo (*Strigops habroptilus*), emu (*Dromaius novaehollandiae*), turkey (*Meleagris spp.*) (*Waite & Taylor, 2014*), bobwhite quail (*Su et al., 2014*), and Japanese quail (*Wilkinson et al., 2016*). The avian GIT is shorter than that of mammals, thus decreasing food transit time (*Golian & Maurice, 1992*), and it is colonized by a unique microbiome adapted to the host (*Stanley, Hughes & Moore, 2014*). Previous research has assessed the impact of chronic low-dose organophosphate insecticide exposure on the GIT microflora of rats and a human GIT simulator (*Joly et al., 2013*). However, to our knowledge, none has studied the impact of organophosphate insecticides on GIT microbiota of any avian species.

In this study, trichlorfon was selected as a representative organophosphate insecticide, and the Japanese quail was chosen as an indicative avian model, as suggested elsewhere (*Foudoulakis et al., 2013*). The general aim of this study was to understand the immediate toxicological effects that trichlorfon residues may produce on the intestinal microbiota of the Japanese quail. Specifically, our aims were to: (i) assess the potential impacts that a single oral dose of a low concentration of trichlorfon (equivalent to that present in a bird when feeding upon arthropods in an orchard 48 h after spraying at the minimum commercially recommended dose) had on the microbial GIT of the Japanese quail after 24 h of exposure. The impact of trichlorfon were assessed by means of shifts in the microbial community composition of the caecum, large intestine and faeces, and (ii) compare these shifts between sexes.

## MATERIALS AND METHODS

### Experimental birds

Fertilized eggs were obtained from Banyard Game Birds farm, Toowoomba, Queensland. The line of quail has been bred for over 20 years without the use of antibiotics and growth promoters. The Banyard Game Birds practice natural hatchery operations without fumigation or sterilization of the eggs, thus ensuring essential transfer of maternal microbiota to the next generation (*Stanley et al., 2013*). Hatchlings were housed together in a rearing pen until they reached three weeks of age, when they were separated to male and female to avoid male fighting over the females, and allocated into a number of cages, each containing four birds, until the start of the experiment at eight weeks of age. The bird stock at the time of the trial was three times higher than the number used in this experiment allowing us to select the birds with near identical initial weights taking each of the birds allocated to the same treatment from the different cage thus avoiding the cage effects on microbiota. At the start of the experiment, birds from the rearing pens were transferred to individual cages and allocated to treatment groups. Throughout the rearing and experimental period the birds were housed in a temperature controlled room at 25 °C, natural light, feed and water was supplied ad libitum. Feed supplied to the birds from hatch

was a commercial turkey starter (Barastoc; Ridley, Victoria, Australia) (22% protein, 2.5% fat, 5% fibre, with added 0.3% salt, 1% calcium, 8 mg/kg copper and 0.3 mg/kg selenium). The starter feed was replaced with Barastoc turkey grower feed (20% protein, 2% fat, 6% fibre, 0.3% salt, 0.95% calcium, 8 mg/kg copper and 0.3 mg/kg selenium) at 4 weeks of age.

## Experimental design

Twenty healthy adult quails of eight weeks-old (10 male and 10 female) were used for this experiment, (average weights; male = 300 g and female = 313.25 g). At the beginning of the experiment, no significant differences in weight were found between treated and untreated quails by sex. Five birds of each sex were assigned to each of two treatments, insecticide-treated and untreated (control). Each of these 20 selected birds were then caged individually throughout the duration of the experiment. Japanese quails are early maturing, short lived birds with the lifespan of up to three years (*Woodard & Abplanalp, 1971*). By six weeks of age all females were laying eggs and all birds, both male and female were fully mature and considered adult.

Trichlorfon residue was estimated using a model specifically designed to calculate pesticide residues on avian food items (i.e., T-REX model, developed by the US Environmental Protection Agency (USEPA) (*Sullivan & Wisk, 2013*)). The T-REX model estimated that 48 h after a single application of Lepidex 500 (containing 500 g/l trichlorfon) at the lowest commercially recommended dose in tree crops (i.e., 100 ml/100 l), the concentration of trichlorfon residues present on arthropods was 26.42 ppm. Following the toxicity test protocol of the USEPA, the latter concentration was individually adjusted to each bird based on its estimated daily food intake (average 133 g/day) and its bodyweight (average 306.6 g) (*USEPA, 1993*). Final trichlorfon concentrations (average 12 μg trichlorfon/g bird weight) were calculated by multiplying the estimated concentration of the trichlorfon residues present on arthropods by the estimated daily food intake of each bird in mg. These final trichlorfon concentrations were adjusted to a final volume of 1ml using distilled water, and given to birds in the 'treated' group by oral gavage in a single dose. Equal volumes of water were given by gavage to the birds in the 'untreated' group. Quails were continuously observed during the course of the experiment using video cameras to monitor their behaviour. None of the birds died or showed abnormal behaviour. Since the aim of the study was to characterize short-term GIT toxicological responses, birds were euthanized 24 h after the administration of trichlorfon by intramuscular overdose of phenobarbitone sodium IP.

Samples from 3 origins (caecum, large intestine and faeces) were obtained from each quail. Luminal specimens were obtained from caecum and large intestine by removing the contents and placing into sterile containers. Faecal samples from each bird were collected within one hour prior to necropsy. All samples were stored at −80 °C for a maximum of one week until DNA was extracted.

## Animal ethics statement

Animal ethics approval for the present project was obtained from the Animal Ethics Committees of the Central Queensland University (permission number A14/03-309).

## Bioinformatics analysis

Total DNA was isolated using the Bioline ISOLATE Faecal DNA Kit (#BIO-52038) according to the manufacturer's protocols. DNA was amplified across the V3–V4 16S rRNA gene region using Q5 DNA polymerase (New England Biolabs). PCR conditions were; 98 °C for 60 s followed by 30 cycles of 98 °C for 10 s, 49 °C for 30 s, 72 °C for 30 s, followed by a final elongation at 72 °C for 10 min. Sequencing was performed on an Illumina MiSeq (2 × 300 bp), based on the method detailed by *Fadrosh et al. (2014)*. The quality filtered sequences were analysed in QIIME 1.8 software (*Caporaso et al., 2010*). Sequences were joined in QIIME using the fastq-join method with zero percent error allowed across the overlapping region. Sequences were demultiplexed retaining only sequences with Phred quality score higher than 20. OTUs were picked using the Uclust algorithm (*Edgar, 2010*) with 97% sequence similarity threshold and inspected for chimeric sequences using Pintail (*Ashelford et al., 2005*). The OTU table was filtered to remove nonbacterial OTUs and OTUs that were of less than 0.01% abundance. Samples with less than 2,000 sequences were removed from the analysis resulting in 51 samples with high quality joined sequences, 25 from control and 26 samples from insecticide treated groups. The filtered table was normalised using QIIME's default cumulative sum scaling (CSS) method (*Paulson et al., 2013*). Additional taxonomic assignment was performed using blastn (*Altschul et al., 1997*) against the NCBI 16S Microbial database. The complete dataset is available on MG-RAST database (http://metagenomics.anl.gov/) under project ID 4693672.3.

## Statistical analysis

Permutational analysis of variance (PERMANOVA) (*Anderson, 2005*) was used to test for differences in microbiota composition at the OTU level. Specifically, three fixed factors were tested: treatment (treated and untreated quails), Sex (male, female), and Origin (caecum, large intestine, and faeces). Normalised data was used to calculate matrices using Bray-Curtis distances. Data was permuted 9,999 times (*Anderson & Braak, 2003*). Canonical Analysis of Principal Coordinates (CAP) was used to visualize these differences in composition. Analysis of Variance (ANOVA) tests were used to test for differences in community diversity, based on Shannon diversity index values at the OTU level. ANOVA was also used to calculate significant differences at higher microbiota taxonomic levels across factors. Additionally, a heat map was created to depict positive and negative correlations between taxa based on Pearson's correlation values. PERMANOVA and CAP were performed with PRIMER software (v.6.0) and PERMANOVA+ software (v.1.0.6). Analyses of community diversity, taxonomic structure, and heat maps were performed in Calypso (v.3.0) (http://bioinfo.qimr.edu.au/calypso/).

## RESULTS

### Microbiota composition

Microbiota composition was significantly affected by Treatment (Pseudo-$F = 3.544$, $P = 0.006$) and Origin (Pseudo-$F = 11.267$, $P \leq 0.001$), but not by Sex (Pseudo-$F = 1.517$,

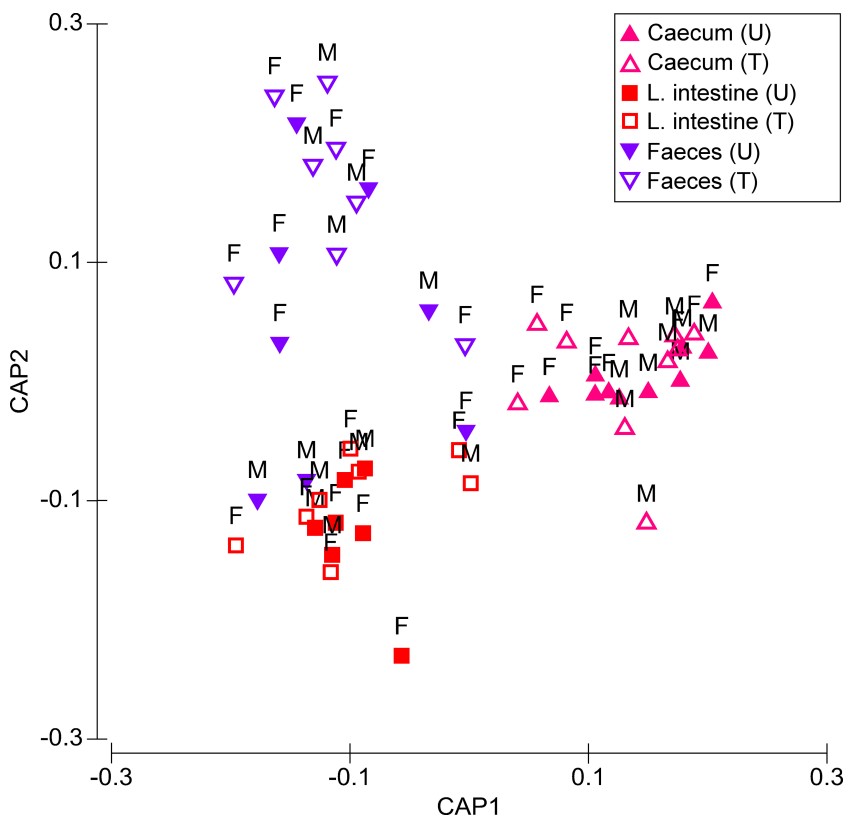

**Figure 1** **Canonical analysis of principal coordinates (CAP) biplot showing a constrained by origin, non-dimensional ordination of microbial community across all factors.** Letter U in the sample legend indicates untreated control while T indicates insecticide-treated birds. M stands for male and F for female birds.

$P = 0.134$) (Fig. 1). There was a significant Treatment × Origin interaction (Pseudo-$F = 2.748$, $P = 0.004$). There were highly significant differences in microbiota composition between treated and untreated quails in caecum ($t = 1.652$, $P = 0.001$), faeces ($t = 1.759$, $P = 0.001$), and large intestine, although the level of significance of the latter was lower than the other two origins ($t = 1.512$, $P = 0.036$). For untreated quails, caecal microbiota differed significantly from those of faeces ($t = 2.883$, $P \leq 0.001$) and large intestine ($t = 3.406$, $P \leq 0.001$). However the large intestine did not significantly differ in microbiota composition from that of faeces ($t = 1.158$, $P = 0.178$) (Fig. 1). In treated birds, differences in microbiota composition between each of the three origins were significant (caecum, $t = 4.037$, $P \leq 0.001$; large intestine, $t = 2.327$, $P \leq 0.001$; faeces, $t = 2.030$, $P \leq 0.001$) (Fig. 1).

Differences between sexes were recorded in the response of microbiota to insecticide. Particularly in the caecum, Treatment and Sex had a significant impact on the microbiota composition (respectively, Pseudo-$F = 2.728$, $P = 0.002$ and Pseudo-$F = 2.139$, $P = 0.023$). Pair-wise tests showed that treated females had a significantly different microbiota composition compared to non-treated females ($t = 1.615$, $P = 0.036$), while in males, these differences were not significant ($t = 1.212$, $P = 0.122$) (Fig. 1). Treatment was the only significant source of variation in both the large intestine (Pseudo-$F = 2.285$,

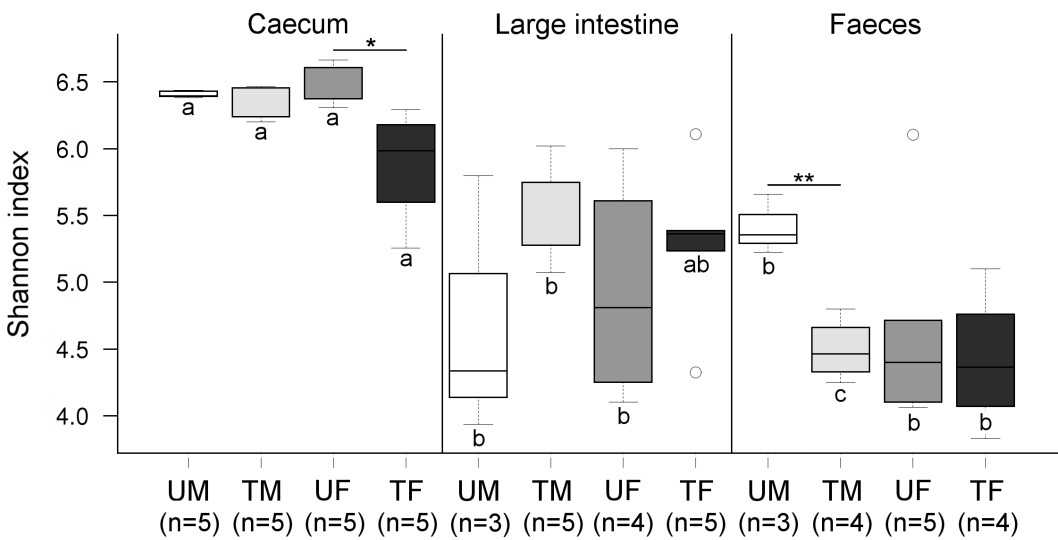

**Figure 2** **Boxplots showing Shannon index values at the OTU level across factors.** Within each origin, significant differences are indicated ('*'$P < 0.05$, '**'$P \leq 0.01$). Same colours indicate groups of quails of the same sex and under the same treatment. Different letters indicate significant differences between these groups across origins. Letter U in the sample legend indicates untreated control while T indicates insecticide-treated birds. M stands for male and F for female birds. Number of samples are indicated.

$P = 0.027$) and faeces (Pseudo-$F = 3.093$, $P = 0.002$) (Fig. 1). However, in the latter origin, pair-wise tests showed that Treatment had a significant effect on the microbiota composition of males (Pseudo-$F = 1.607$, $P = 0.028$), but not on that of females (Pseudo-$F = 1.224$, $P = 0.085$).

## Microbiota diversity

Community diversity differed significantly across each of the three factors ($P \leq 0.001$) (Fig. 2). Overall, the average caecal microbiota diversity, at an OTU level, was higher than the other two origins. Microbiota diversity in the caecum differed significantly between treated and untreated females ($P = 0.019$), but no significant differences were shown for the male microbiotas ($P = 0.185$) (Fig. 2). Diversity in the large intestine did not differ significantly across Treatment or Sex (Fig. 2). In faeces, significant differences were found between treated and untreated males ($P = 0.003$), but not for the females ($P = 0.605$) (Fig. 2).

## Caecum

Overall, the caecum microbiota was represented by 9 phyla, 27 orders, 56 families, 87 genera, and 1,138 OTUs, based on 97% sequence similarity and using the Uclust algorithm. Across factors, the main three phyla ordered by relative abundance levels were Firmicutes (constituting approximately 81% of abundance across groups), Bacteroidetes (10%), and Proteobacteria (8%). The main genera represented in the caecal microbiota were *Ruminococcus* (15%), *Bacteroides* (10%), *Faecalibacterium* (8%), *Clostridium* (6%), and *Eubacterium* (4%).

Comparisons of the microbiota at the phylum level showed that in treated quails the abundance of Proteobacteria was significantly increased ($P = 0.007$) (Fig. 3A). This

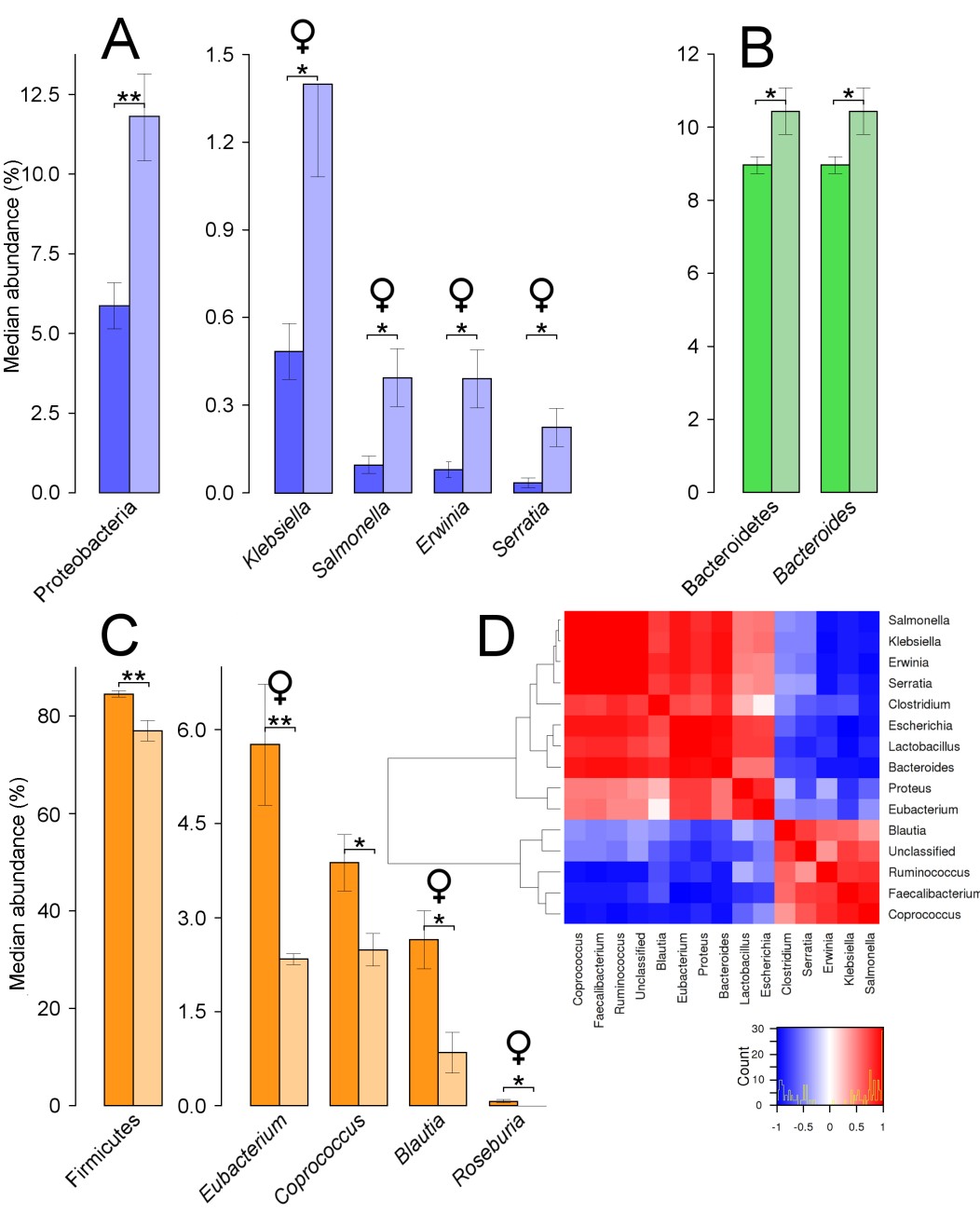

**Figure 3 Caecal microbial taxa showing significant shifts.** These taxa were grouped in the phyla Proteobacteria (A), Bacteroidetes (B), and Firmicutes (C); '*' indicates $P \leq 0.05$, and '**' $P \leq 0.01$. Light colours indicate treated birds. Sex-dependent significant shifts are indicated with symbols. To facilitate comparisons across origins, taxa in the same phylum is indicated by same colours in Figs. 3–5. (D) shows correlations between genera in treated females (at levels of abundance > 0.4%), based on Pearson's correlation. A histogram indicates the distribution of counts based on the correlation values.

occurred due to changes at the family level, by similar significant relative increases in abundance of Enterobacteriaceae (5.4–9.7%, $P = 0.009$). Four genera contained in this family increased significantly in abundance within the treated females, *Klebsiella* ($P = 0.016$), *Salmonella* ($P = 0.032$), *Erwinia* ($P = 0.016$), and *Serratia* ($P = 0.021$) (Fig. 3A). Nevertheless, each of the latter three genera represented relative abundances below 1%. Significant increases also occurred in the phylum Bacteroidetes ($P = 0.043$), linked to equally significant increases in the genus *Bacteroides* (Fig. 3B). The relative abundance of the phylum Firmicutes decreased significantly from 84% in untreated to 77% in treated quails ($P = 0.005$). These results were related to a significant decrease of the family Lachnospiraceae (55–47%, $P = 0.023$). Three genera present in this family significantly decreased in treated females, *Eubacterium* ($P = 0.008$), *Blautia* ($P = 0.032$), and *Roseburia* ($P = 0.025$) (Fig. 3C). When combining male and female quails, *Coprococcus*, an additional genus from the same family, decreased significantly ($P = 0.035$) (Fig. 3C). No significant differences were found at the genus level in males. The heat map showed how each of the genera increasing significantly in female quails were positively correlated between each other and negatively correlated with four other genera: *Coprococcus*, *Faecalibacterium*, *Ruminococcus*, and *Blautia* (Fig. 3D).

## Large intestine

The overall composition of the large intestine microbiota was represented by a total of 10 phyla, 33 orders, 71 families, 103 genera, and 1,068 OTUs. Based on averaged values across all factors, the main phyla were Firmicutes (65%), Proteobacteria (21%), Bacteroidetes (7%), and Actinobacteria (5%). The five most abundant genera were *Lactobacillus* (25%), *Ruminococcus* (6%), *Bacteroides* (6%), *Clostridium* (4%), and *Faecalibacterium* (3%).

Based on ANOVA rank tests, there were significant differences at the phylum level, as the abundance of Firmicutes decreased from 76% in untreated to 55% in treated quails ($P = 0.014$) (Fig. 4A). These changes were linked to significant decreases at the genus level in *Lactobacillus* ($P = 0.029$) (Fig. 4A). The most abundant species under this genus was *L. agilis*, which was assigned, using blast against 16S Microbial database, to a total of 36 OTUs as 'first choice' identity (average = 93.3%, min = 86%, max = 98%). This species' abundance dropped significantly from 25% in untreated to 5% in treated quails ($P = 0.014$) (Fig. 4A). The phylum Verrucomicrobia, with a relative abundance of less than 1%, significantly increased in treated quails ($P = 0.038$), due to equally significant increases in the genus *Akkermansia* (Fig. 4B). Although not-significant, the Proteobacteria phylum increased from 15% in untreated to 27% in treated quails, partly due to significant increases at the genus level in *Klebsiella* ($P = 0.019$), and at lower proportions (i.e., <1%) in *Rhodospirillum* ($P = 0.045$), *Alkanindiges* ($P = 0.019$) and *Lysobacter* ($P = 0.045$). Also, significant increases in the genus *Agrobacterium* ($P = 0.044$) (comprised of an OTU 100% identical to *Agrobacterium fabrum*) were found only in females (Fig. 4C). Lastly, the proportion of unclassified genera increased significantly in treated quails ($P = 0.029$) (Fig. 4D).

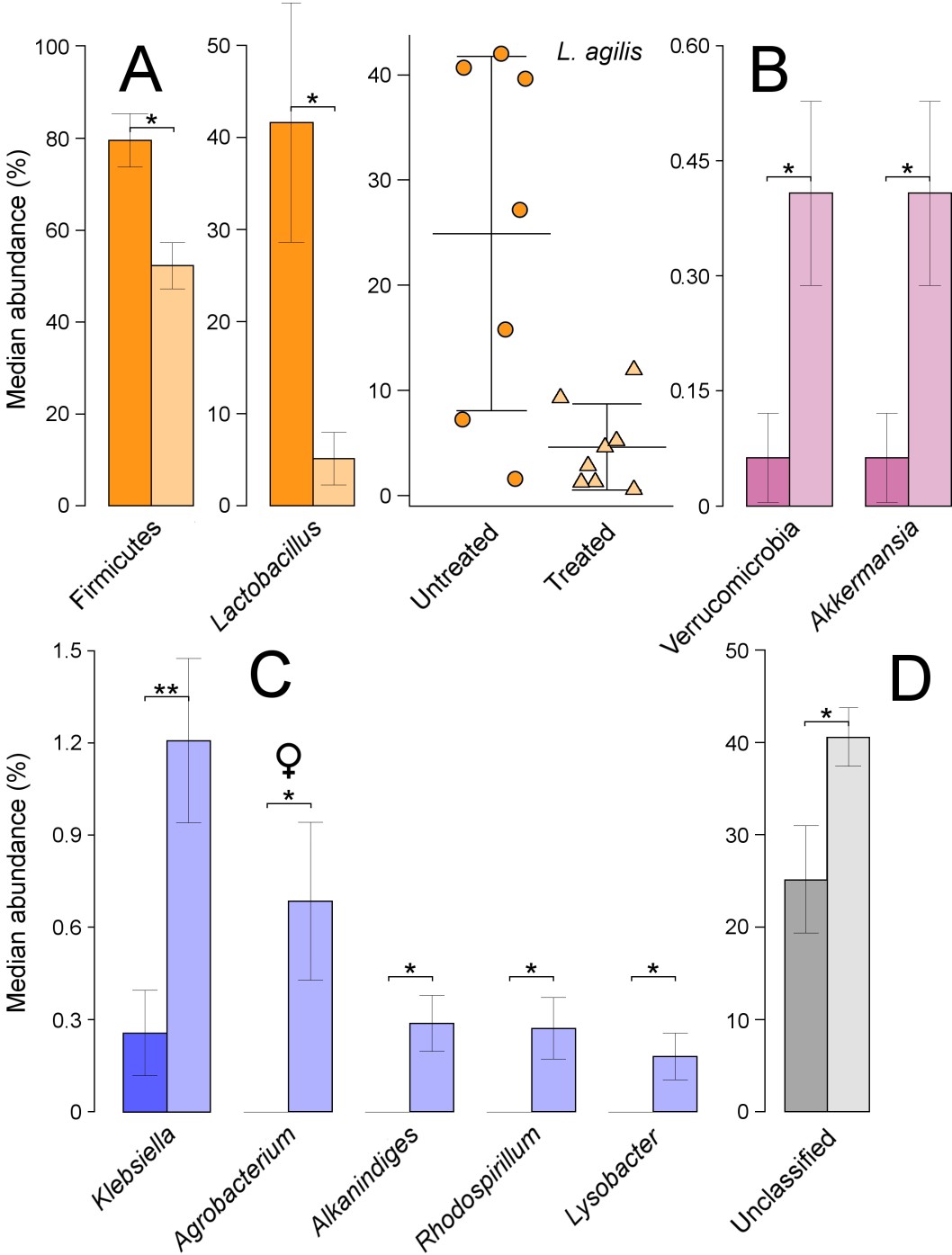

**Figure 4 Microbial taxa in the large intestine showing significant shifts.** These taxa were grouped in the phyla Firmicutes (A), Verrucomicrobia (B) and Proteobacteria (C). Unclassified taxa is shown in panel D; '*' indicates $P \leq 0.05$, and '**' $P \leq 0.01$. Light colours indicate treated birds. Sex-dependent significant shifts are indicated with symbols. To facilitate comparisons across origins, taxa in the same phylum is indicated by same colours in Figs. 3–5.

Additionally, the phylum Actinobacteria increased non-significantly from approximately 3% in untreated to 7% in treated quails, partly because of relatively low but significant increases in *Janibacter* (0–0.3%, $P = 0.007$), and only in females, in *Rothia* (0–0.5%, $P = 0.044$). Additionally, treated females experienced significant increases in the family Bacillaceae (0–0.7%, $P = 0.044$).

## Faeces

Overall, after initial OTU filtering described previously, microbial taxa in quail faeces were classified into 9 phyla, 31 orders, 67 families, 99 genera, and 980 OTUs. Across factors, the main phyla were Firmicutes (approximately 57% of abundance), Proteobacteria (25%), and Bacteroidetes (8%), and the main genera were *Lactobacillus* (10%), *Faecalibacterium* (6%), and *Bacteroides*, *Clostridium*, and *Ruminococcus* (each 4%).

At the phylum level, only treated males showed significant increases in Verrucomicrobia ($P = 0.05$), which were linked to equally significant increases in the family Verrucomicrobiaceae, and the genus *Akkermansia* (Fig. 5A). This genus was composed of a single OTU which was assigned to the species, *A. muciniphila*, at 100% of identity across the amplified region. Furthermore, the phylum Firmicutes decreased non-significantly when combining results from both sexes, from 59% in untreated to 55% in treated quails. Two families from this phylum showed significant increases, Turicibacteraceae ($P = 0.002$), and Streptococcaceae ($P = 0.001$), the latter partly caused by a significant increase in the genus *Streptococcus* ($P = 0.001$) (Fig. 5B). Other genera from the same phylum showed significant shifts: significant decreases were found in *Eubacterium* ($P = 0.032$) in treated males, and significant increases were found in *Jeotgalicoccus* ($P = 0.042$) in treated females (Fig. 5B). Sphingomonadaceae, a family from the phylum Proteobacteria, was significantly increased in treated quails (0.33–1.20% $P = 0.018$), which resulted from significant increases in the genus *Sphingomonas* ($P = 0.010$) (Fig. 5C). Two additional genera were significantly increased in abundances in treated quails, *Delftia* ($P = 0.010$) and *Brevundimonas* ($P = 0.031$), while *Escherichia* showed a significant decrease ($P = 0.005$) (Fig. 5C). Prevotellaceae, a family from the phylum Bacteroidetes, showed significant increases ($P = 0.004$), caused by an equally significant increase in the genus *Prevotella* (Fig. 5D). Additionally, two genera, each from different phylum, varied significantly their abundances, *Allobaculum* (Tenericutes) ($P = 0.027$), and *Micrococcus* (Actinobacteria) ($P = 0.050$), the latter only in males (Fig. 5E).

Additionally, several significant changes in abundance occurred in phylotypes that were present in low abundance (below 2%). There was a significant increase in treated quails in the family Bifidobacteriaceae (0.2–0.8%, $P = 0.029$), linked to equally significant increases in the genus *Bifidobacterium*. At similar low abundances, there were significant increases in the family Pasteurellaceae (0–0.6%, $P = 0.05$) linked to equally significant increases in the genus *Haemophilus* (0–0.6%, $P = 0.03$). There were also significant increases in the families Clostridiaceae (0.1–1.1%, $P = 0.016$), Propionibacteriaceae (0.4–1.2, $P = 0.022$), and 'Clostridiales Family XI' (0–0.6%, $P = 0.013$). Increased abundance of the latter two families were linked to an equally significant increase in the genera *Propionibacterium* and *Anaerococcus*. Additionally, there were significant increases in the genus *Oceanobacillus*

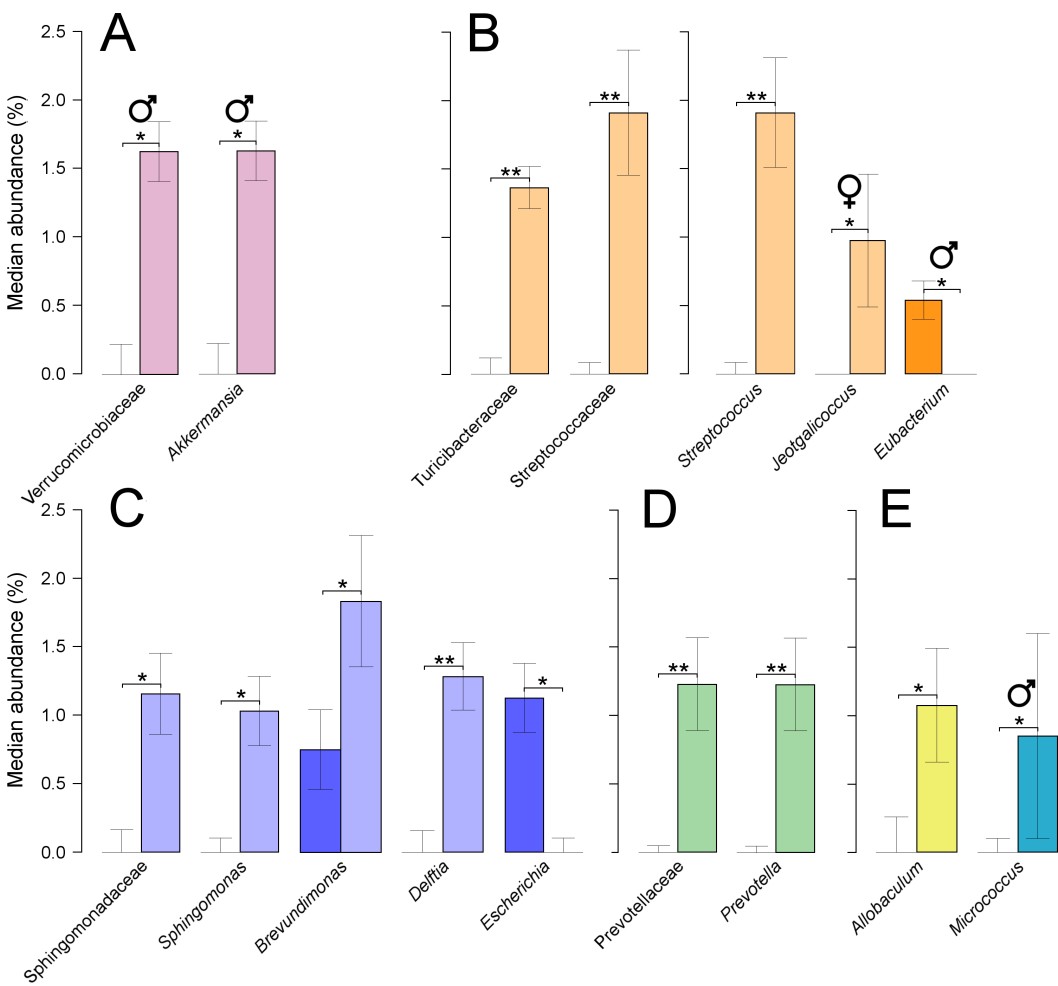

**Figure 5 Faecal microbial taxa showing significant shifts.** These taxa were grouped in the phyla Verrucomicrobia (A), Firmicutes (B), Proteobacteria (C), Bacteroidetes (D), and shown together, Tenericutes and Actinobacteria (E). '*' indicates $P \leq 0.05$, and '**' $P \leq 0.01$. Light colours indicate treated birds. Sex-dependent significant shifts are indicated with symbols. To facilitate comparisons across origins, taxa in the same phylum is indicated by same colours in Figs. 3–5.

(0–0.6%, $P = 0.013$) and significant decreases in the genus *Eggerthella* (0.25–0%, $P = 0.013$).

## DISCUSSION

This work explored the short-term impacts of an organophosphate insecticide on the intestinal microbiota of the Japanese quail. Few studies have assessed the impact of insecticides on the GIT (e.g., *Joly et al., 2013*), but to our knowledge none has been conducted on avian systems in spite of results highlighting the central role that the GIT resident microflora plays in the bird's health (*Stanley et al., 2012*; *Van Dongen et al., 2013*). Our results revealed that a single oral dose of trichlorfon at low-concentration induced significant shifts in bacterial community composition and diversity across origins. Particularly, our results showed overall increases in Proteobacteria, which have been

recently regarded as an indicator of microbial dysbiosis in mammals (*Shin, Whon & Bae, 2015*). In humans, microbiota dysbiosis has been associated with immunological dysregulation (*Round & Mazmanian, 2009*) and diseases such as diabetes (*Qin et al., 2012*), irritable bowel syndrome (*Chassard et al., 2012*), colorectal cancer (*Sobhani et al., 2011*), and polycystic ovarian syndrome (*Tremellen & Pearce, 2012*). Although the role of the GIT microbiota is not yet fully understood in humans nor in birds, high microbial diversity is generally considered an indicator of a healthy balance. In contrast, an imbalance generally promotes overgrowth of pathogenic or non-beneficial microbiota, and diseases may occur (*Karlsson et al., 2010*). This study showed consistency in microbial shifts at high taxonomic level in caecum and large intestine, but not on faeces. However, at lower taxonomic levels, contrasting microbial responses were found between all treatments and origins.

## Caecum

Caecal microbiota showed significantly higher diversity values at an OTU level and highly significant differences in composition compared with the other two origins. These results agree with findings from other studies of Japanese quail (*Wilkinson et al., 2016*). Significant shifts in microbiota were observed at the phylum level. Firmicutes numbers decreased, while gram-negative phyla such as Proteobacteria and Bacteroidetes increased in treated quails. Previous studies have shown similar shifts in these three phyla, associated to applications of synthetic pyrethroid insecticides in plant tissue (*Zhang et al., 2009*).

Interestingly, our results showed a significant effect of sex on the microbiota composition in the caecum, which did not occur in the other two origins. Only in females, five genera belonging to Proteobacteria and Bacteroidetes were increased significantly in abundances (i.e., *Salmonella*, *Klebsiella*, *Erwinia*, *Serratia*, and *Bacteroides*). It is possible that these and other observed increases in microbiota abundance resulted from their tolerance to the insecticide. Some bacteria have the catabolic ability to dissolve insecticides and use them as a carbon source, an ability termed as biodegradation (*Akbar, Sultan & Kertesz, 2014*). For instance, the latter study showed that Proteobacteria was the only dominant bacterial group selected by enrichment culturing using chlorpyrifos (and organophosphate pesticide) as the carbon source. Other studies have found that *Klebsiella* and *Serratia* are efficient organophosphate biodegraders (*Cycoń, Wójcik & Piotrowska-Seget, 2009*; *Pino & Peñuela, 2011*; *Sasikala et al., 2012*). Furthermore, *Joly et al. (2013)* found significant increases in *Bacteroidetes* in both the intestine of rats and in a human intestine simulator, after the continuous application of low doses of an organophosphate insecticide. *Burkholder et al. (2008)* showed that acute stressors resulted in shifts in GIT microbiota and increases in *Salmonella* abundances. Thus, it is likely that some of the aforementioned taxa increased as they were not adversely affected and hence had a relative advantage compared to other bacteria, as a consequence of their biodegradation ability. Additionally, three genera from the Lachnospiraceae family decreased, but *Blautia* and *Eubacterium* did so only in females. *Eubacterium* produce butyrate, which is considered a gut health promoter through cell growth and colonic cell turnover (*Gong et al., 2002*). Another study reported increases in abundance of a number of caecal bacteria, including *Blautia*, associated to an improvement of 10% in feed conversion ratio in broilers (*Li et al., 2016*). In general, the Lachnospiraceae

family has been associated with improved growth performance in chickens (i.e., lower food conversion ratio) (*Stanley et al., 2012*). Therefore, the aforementioned decreases are potentially associated with negative effects on the health of quails.

Other studies have indicated that there are sex-dependent changes in environmental conditions (e.g., pH and oxygen gradients, bile acid levels) throughout the GIT of female quails (*Flint et al., 2012*; *Islam et al., 2011*) and this may provide a clue as to why there are sex related differences in the microbiota within the caecum. Our results indicate that in the caecum of treated females, a number of bacteria, including all Enterobacteriaceae members, formed a microbial consortium which negatively correlated with the abundance of Firmicutes. Although we cannot predict the health risks that these microbiota shifts may invoke, a number of studies have shown the detrimental impact that some of these bacterial groups have. Most of the genera that increased in abundance belonged to Enterobacteriaceae. Based on findings from deceased birds, it has been suggested that bacteria from this family predispose birds to diseases (*Glünder, 2002*). Particularly in the *Salmonella* genus, several reports in Europe and America have associated the death of different wild bird species due to Salmonellosis (*Craven et al., 2000*). Other studies associated very high abundances of Bacteroidetes in patients suffering irritable bowel syndrome (*Ng et al., 2013*). Moreover, increases in *Bacteroides* growth in the intestines of two different animals were associated with decreases in probiotics such as lactobacilli and bifidobacteria (*Joly et al., 2013*). Results from these studies suggest that the shifts we observed in our study, particularly in the case of female quails, could predispose birds to the development of diseases. However, further research is necessary to clarify the mechanisms and the consequences of these microbiota shifts.

## Large intestine

Significant Treatment × Origin interactions in microbiota composition were found in this study. Significant changes in microbiota composition were found between the large intestine and faeces in treated quails while no significant changes were found in the untreated quails. There were large reductions of *Lactobacillus* (phylum Firmicutes) in the large intestine of treated birds. This genus is generally present in normal mucosal microbiota in most animals (*Kapczynski, Meinersmann & Lee, 2000*), and numerous studies have investigated their probiotic effects in avian systems. Early research showed that high abundance of this genus can inhibit *Escherichia coli* growth in chicken crop (*Fuller, 1977*). More recently, *Timmerman et al. (2006)* showed that administering probiotic *Lactobacillus* species to broilers promoted their growth and reduced mortality. In our study, the most affected species of *Lactobacillus* was *L. agilis*. *Lan, Sakamoto & Benno (2004)* showed that this species, along with other *Lactobacillus* strains, restored the balance and maintained the natural stability of bacterial microbiota after heat stress-induced changes. In another study, probiotic supplements composed of *L. agilis* and *L. salivarius* enhanced lactobacilli and bifidobacteria abundance, and inhibited *Salmonella* growth in a simulated chicken caecum (*Meimandipour et al., 2010*). However, there are number of studies questioning the beneficial effects of *Lactobacillus* probiotics in birds (e.g., *Goodling, Cerniglia & Hebert, 1987*) and their effect on the bird health cannot be taken as independent from the rest of the

microbial community structure as it is often done in the literature. In this study, differences observed in *Lactobacillus* abundance between treated and untreated quails could point to the lack of adaptive responses of this biota to the new chemical environment of the large intestine. Additionally, it is possible that the reductions observed in *Lactobacillus* in treated quails could cause detrimental consequences for their health.

Similarly to the results found in caecum, the large intestine of treated quails had increases in the abundance of Proteobacteria, which occurred as a result of significant increases in five genera (i.e., *Alkanindiges*, *Lysobacter*, *Agrobacterium*, *Rhodospirillum*, and *Klebsiella*). Significant increases in the latter genus were also found in the caecum. Furthermore, treated quails showed increased abundances of Verrucomicrobia, a poorly characterized phylum (*Hou et al., 2008*), both in the large intestine and in faeces. Increases in this phylum were attributed to a sole species, *Akkermansia muciniphila*. This bacteria degrades intestinal mucin to obtain carbon and nitrogen (*Van Passel et al., 2011*). In mice, it was suggested that intestinal bacteria producing excessive mucin degradation could contribute to inflammatory bowel diseases, since of luminal antigens to the intestinal immune system was facilitated (*Ganesh et al., 2013*). Increases in *A. muciniphila* abundance have been negatively correlated with body weight in humans and rodents (*Everard et al., 2013*). Moreover, studies have shown that overweighted people undergoing a fasting program showed increased abundance of *Akkermansia* in their gut microbiota (*Remely et al., 2015*). During our study, insecticide-treated quails were fasting (by reducing feed intake), most likely as a consequence of trichlorfon ingestion. It is thus hypothesised that fasting in insecticide-treated birds might have contributed to higher abundances of *A. muciniphila*, in comparison with untreated birds which were feeding normally. Lastly, our results showed further insecticide impact through an increase of unclassified bacteria to relative abundances above 40% in treated quails. Although the identity of the latter group is unknown, it is possible that it could be partially represented by phyla which showed positive responses to insecticide (e.g., Proteobacteria and Bacteroidetes), as observed in other origins.

## Faeces

Several microbiota taxa were significantly affected by trichlorfon in faeces, however, most of these changes occurred at much lower abundance levels than in the other two origins. Compared to the other origins, faecal microbiota showed consistent responses in taxa associated to the phyla Verrucomicrobia and Proteobacteria as shown in this study. However, increases in taxa in the Firmicutes phylum occurred concurrently with decreases in the same phylum in the other origins, showing an inconsistency in the faecal microbiota responses. It is not unusual to observe inconsistencies such as a lack of a unidirectional, dose–response correlation in microbial toxicology tests (*Petersen, Dahllof & Nielsen, 2004*; *Widenfalk et al., 2008*). Furthermore, it is well accepted that avian faecal microbiota oscillates based on the regular emptying of different GIT sections (*Sekelja et al., 2012*). In a recent study, source–sink predictions of the seeding dynamics across different GIT sections of Japanese quail showed that most of the faecal microbiota composition had a multi-source predicted origin, with major origins contributing to it being caecum, and

large intestine (*Wilkinson et al., 2016*). Thus, it is possible that the results observed in the faecal samples of treated quails originated from a combined reaction of different origins in the GIT to the insecticide.

## CONCLUSIONS

A single sub-lethal exposure to an organophosphate insecticide, at concentrations similar to levels likely to be ingested following insecticide spraying in an orchard, caused significant changes in avian gut microbiota within 24 h. The effect size of this change, and the shifts in microbial groupings, are consistent with microbiotas that previous studies have shown may predispose birds to diseases. The changes differ between parts of the GIT, and interestingly also differ between sexes, particularly in the caecum.

The study revealed that increases in abundance of certain microbial taxa occurred as a result of direct or indirect positive interactions with trichlorfon administration, some possibly as a result of their biodegradation ability. On the other hand, reductions observed in other groups likely arose from their susceptibility to the insecticide, which may have produced antagonism and/or competition with overgrowing biodegradative and/or insecticide-tolerant taxa.

This study raise several questions that should be addressed to understand the interactions between GIT microbiota and avian hosts, the effects of pesticides on these interactions, and the overall consequences for the bird's health. Further research is necessary to unravel the mechanisms driving the interactions, and to understand how these are potentially associated with damage, and/or the development of diseases in the GIT. Additionally, further study of short term physiological responses and longer term health impacts are warranted based on the study results.

Considering that we carried out this study in a laboratory under defined and stable conditions, our results can only partially mimic what is likely to occur under field conditions after insecticides are sprayed. Nevertheless, our results suggest that changes in microbiota associated to even small doses of insecticide could impose health risks on birds, particularly in females. The implications of the significant response to a low dose of an organophosphate insecticide are that acute and chronic health impacts are likely to occur in avian populations if similar GIT responses occur under field conditions. These results are relevant for the conservation of avian communities in agricultural areas where organophosphates are regularly sprayed.

## ACKNOWLEDGEMENTS

We are grateful to Clive and Erika Wiley (owners of Banyard Game Farms), who provided the quails used in this study. We would like to thank Honglei Chen for operating the Illumina MiSeq instrument. All data analyses were performed on the Isaac Newton High Performance Computing System at Central Queensland University. We acknowledge the support received from Jason Bell in all aspects of high-performance computing as well as the additional support received from Giselle Weegenaar, Ingrid Christiansen and Judy Couper.

### Funding

During this study, Eduardo Crisol-Martínez was supported by an Australian Postgraduate Award and an International Postgraduate Research Award. Dragana Stanley is an Australian Research Council (DECRA) Fellow. Poultry CRC supported Ngare Wilkinson. The funders had no role in study design, data collection and analysis, decision to publish, or preparation of the manuscript.

### Grant Disclosures

The following grant information was disclosed by the authors:
Australian Postgraduate Award.
International Postgraduate Research Award.
Poultry CRC.

### Competing Interests

The authors declare there are no competing interests.

### Author Contributions

- Eduardo Crisol-Martínez conceived and designed the experiments, performed the experiments, analyzed the data, wrote the paper, prepared figures and/or tables, reviewed drafts of the paper.
- Laura T. Moreno-Moyano, Ngare Wilkinson and Tanka Prasai performed the experiments, reviewed drafts of the paper.
- Philip H. Brown conceived and designed the experiments, reviewed drafts of the paper.
- Robert J. Moore conceived and designed the experiments, performed the experiments, contributed reagents/materials/analysis tools, reviewed drafts of the paper.
- Dragana Stanley conceived and designed the experiments, performed the experiments, analyzed the data, contributed reagents/materials/analysis tools, reviewed drafts of the paper.

### Animal Ethics

The following information was supplied relating to ethical approvals (i.e., approving body and any reference numbers):

Animal ethics approval for the present project was obtained from the Animal Ethics Committees of the Central Queensland University (permission number A14/03-309).

### Patent Disclosures

The following patent dependencies were disclosed by the authors:
Bioline ISOLATE Faecal DNA Kit (#BIO-52038)

### Data Availability

The complete dataset is available on MG-RAST database (http://metagenomics.anl.gov/) under project ID 4693672.3.

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
