# Peer review of "A low dose of an organophosphate insecticide causes dysbiosis and sex-dependent responses in the intestinal microbiota of the Japanese quail (Coturnix japonica)"

_PeerJ, doi:10.7717/peerj.2002_

## Round 0.1 · original submission · Major Revisions

· Academic Editor

Major Revisions

All reviewers agreed in their support of your study as an important and interesting contribution to elucidating the effects that pesticide residues can have on wildlife health. Nevertheless, they also make a number of detailed suggestion in order to deal with important issues regarding experimental design, data analysis and data interpretation.

Reviewer 1 ·

Basic reporting

The article is well written and includes sufficient background in order to comprehend the goal of the experiment.
Figures are clear and all data relevant to the hypothesis are included.

Experimental design

The research question is clearly defined and methods are described in detail.

Validity of the findings

Statistical analysis has been performed and the conclusions made are connected to the inital investigation question

Additional comments

Some minor remarks
line 110: include the use 'of' antibiotics
line 113: grouped 4 to a cage... this is unclear to me
line 115: change ab into ad
line 121: what is the age of an adult quail?
line 132: it is not clear how the 12µg of trichlorfon was calculated
line 142: can the faecal samples be linked to the bird? Clarify this
plural of bacterium is bacteria: change accordingly
change along the tekst: impact on microbiota instead of in microbiota

·

Basic reporting

Overall, the manuscript is well written and sufficient background information is provided to support questions addressed by this work. However, few changes and corrections throughout the text should be made to improve readability. Authors should provide accession numbers for the sequencing raw data.

Experimental design

Research questions addressed are relevant and clearly defined. The methodology described cannot be completely reproduced by another investigator, therefore, it needs to be expended (see general comments). Bioinformatics and statistical analyses seem to be correctly conducted, however, I do have some concerns regarding the methods and approaches used (see general comments).

Validity of the findings

The manuscript shows that an exposure, via ingestion of contaminated food, to a sub-lethal dose of an organophosphate pesticide causes significant shifts on the gut microbiota of the japanese quail. Authors discuss and speculate about the potential impacts that such shifts might have on the health of birds. However, there was no treatment to assess recovery after exposure, which could have indicated to which extent the exposure poses a treat to the health of these birds. Furthermore, there is no discussion about the role of the microbial groups that are changing in the GIT. I am not, by any means, an expert in gut microbiota, but a shift in taxonomic groups not necessarily means a shift in function. Perhaps, the author could try to fill this gap by analysing their 16S rRNA data with PICRUST.

Additional comments

Introduction
L64-73: what are the concentrations that are representative of exposure levels on the filed (environmental relevant concentrations)?
L77: Are birds exposed to pesticides only by food ingestion?
L88: "...most of the studies characterising the GIT microbiota" of avian species "have been carried out" in chicken models. (suggestion)
L103: define low, taking into account environmental relevant concentrations.

Material and Methods
L115: ad libitum, rather than ab libitum???
L121-L125: what was the age of the birds at the beginning of the experiment? Definitely more than 4 weeks. What is the life span of these birds? In the introduction authors mention that GIT microbiota play an important role in the development and maturation of the immune systems. Is this relevant for these birds?
L126-130: Authors used a model to estimate pesticide residue of avian food sources, however, nothing is mentioned to indicate that the concentration of pesticide used to estimate the residue is the concentration normally applied of the field.
L149-150: Conditions used for PCR relations are not indicated.
L157: what were the non bacterial OTUs, archaea?
L162: what was the reason for choosing the NCBI Microbial database for taxonomy assignment? It seems to me that authors could have chosen a more appropriate database like SILVA.
L164: Provide accession numbers.
L170: is 9999 times correct? Or it was used 999 times?
L166-178: No singletons and doubletons were discarded. This might have a significant effect on the analyses performed, especially because authors performed all the analyses on OTU level.
L178: I was not able to access the webpage provided as source of the calypso software.

Results
The results section is rather descriptive and many of the results indicating shifts on the GIT microbiota are presented on genus or species level. I would be very careful to base my results on such lower levels. It is well know that OTU clustering algorithms are far from perfect and also that erros introduced during the preparation and sequencing of the samples might have an impact on the diverse and taxa observed. Moreover, some of the results presented show changes on taxa that are present in a very small relative abundance. Would the discard of singletons and doubletons change the observed results?

The substitution of the word proportional by relative might improve readability (example L227, 233, 237, etc.).

There are sentences that should be rephrased, since they are too long and difficult to understand. Check L271-275 as an example.

Discussion
L355-357: Include examples of effects of organophosphate insecticides in bacteria in other environments. Are the same phyla affected?
L372-374: Do bacteria need to be able to degrade the insecticide to have an advantage?
L373: should be bacteria, rather than bacterium.
L386: bacteria, rather than bacterium (check text throughly).
L438: include references.
L438-440: What could be an explanation for the increase of Firmicutes in the faeces material, while in the other origins this phylum decreases significantly.
L446-447: sentence seems to be incomplete. Also doesn't this sentence contradict the sentence on L438-440?

Discussion
L451-453: suggestion: A single sub-lethal exposure to an organophosphate insecticide, at concentrations similar to levels likely to be ingested following insecticide spraying in an orchard, caused significant changes in avian gut microbiota within 24 hours.
L453: The size of this change, it reads/sounds strange.
L457-461: Check comment on L372-374. I dont believe that all bacteria that responded positively to the exposure did so because they are able to degrade the insecticide. Moreover, reduction of bacterial groups could have happened not only by susceptibility to the insecticide. An indirect effect does not mean susceptibility to the insecticide.
L462: which questions?

·

Basic reporting

Feed formula as well as feed ingredients may have an enormous impact on intestinal microbiota composition. The ingredients used to formulate the feeds as well as the percentages included should be reported. On line 117, the impression is given as if no premix of vitamins and minerals was used, but only sodium, copper, calcium and selenium was added.

Experimental design

There are several flaws in the exeperimental set up. On line 113, it appears as if birds were housed 4 per cage. With 20 birds, that means 5 cages, thus not evenly distributed over the 2 treatments. It is also unclear whether the males and females were housed separately. It is well established that cage effects on microbiota composition can be very important. It is unclear why fecal samples were used. Moreover, it is well known that birds produce two types of droppings, cecal droppings and colonic droppings. the former are much more sticky and thus can be distinguished from the latter. It is unclear which type of droppings were used here. Moreover, it is inclear how long after defecation the feces were collected. this may have considerable influence on the microbial composition.

Validity of the findings

the data may be biased by cage effects, distribution of the sexes over the cages, the limited numbers of animals (2 groups of 5 for each sex). In the discussion the authors (mis)use literature data. for instance; when refering to the paper of Li et al., 2016) they use the Blautia argument, but they dismiss that Li et al. also found an increase of Desulfovibrio associated with improved FCR, which is in conflict with their findings and with their reference to Shin et al (2015). Moreover, the paper of Shin et al. relates to mammals, not birds. another example: there are just as many studies showing that Lactobacillus probiotics have no beneficial effect. there are studies showing that Akkermansia is enriched in studies showing beneficial effects of prebiotics.

Additional comments

Ideally, the analyses of microbiota should have been done on the same animals before and after treatment. there are conflicting reports whether faeces are representative or not for intestinal microbiota.

---

## Round 0.2 · accepted · Accept

· Academic Editor

Accept

The much improved manuscript has been evaluated by myself and one of the reviewers of the initial submission, and we both felt that you very well responded to all issues raised by reviewers initially.

·

Basic reporting

The authors have responded correctly to all my remarks

Experimental design

The authors now have described their experimental design correctly. The design is indeed O.K.

Validity of the findings

The findings are useful for researchers working in this field